# Prenatal and early postnatal periods differentially shape the maturation of human cortical microstructure and myelin

Thanos Tsigaras[1,2], Juergen Dukart[1,2], Stuart Oldham[3,4], Simon B. Eickhoff[1,2], Casey Paquola [1]*

1 Institute for Neuroscience and Medicine (INM-7), Forschungszentrum Jülich, Jülich, Germany,
2 Institute for Systems Neuroscience, Heinrich Heine University Düsseldorf, Düsseldorf, Germany,
3 Developmental Imaging, Murdoch Children's Research Institute, Parkville, Australia, 4 Turner Institute for Brain and Mental Health, Monash University, Melbourne, Australia

* c.paquola@fz-juelich.de

## Abstract

During late gestation and early postnatal development a combination of intrinsic and extrinsic factors drive the maturation of the human cortex. This process is regionally heterogeneous, with cortical areas developing at different paces and trajectories. Leveraging submillimetre T1-weighted/T2w-weighted (T1w/T2w) magnetic resonance imaging (MRI) from pre- and full-term neonates ($n = 599$, 26–44 weeks), we sampled intracortical profiles across the cortex and characterized the profiles' shapes according to their central moments. We found that gestational age at birth dominated the effects on early cortical development, with significant, global increases in intracortical homogeneity and a bimodal change in the balance of myelin-sensitive signal between superficial and deeper cortical layers. On the other hand, weeks since birth (i.e., postnatal age) exhibited different effects on myelin, with increasing intracortical heterogeneity and intracortical balance only shifting towards deeper layers in posterior temporal, occipital, medial parietal areas and some prefrontal areas. These effects align with low spatial-frequency geometric eigenmodes of the human cortex, specifically the anterior–posterior and superior-inferior axes. Our findings demonstrate that separating prenatal from postnatal influences, and analyzing intracortical profiles rather than macroscale features, provides finer-grained insights into how human cortical myelin changes during perinatal development and lays the groundwork for investigating the biological underpinnings that govern normative cortical maturation.

## Introduction

Perinatal neurodevelopment is a multifaceted procedure, comprising the early stages of the molecular, structural, and functional maturation of the infantile human cortex. While non-human animal studies and human postmortem research have laid out the

**Data availability statement:** Data were obtained from the developing Human Connectome Project (https://www.developing-connectome.org/), which is openly available to download for research purposes via the NIMH data archive (https://nda.nih.gov/edit_collec-tion.html?id=3955). Code and additional data for this study are openly available at a Github repository (https://github.com/ThanosTsig/InfantMicro) which is archived in Zenodo (https://doi.org/10.5281/zenodo.18910237).

**Funding:** This work was supported by the Deutsche Forschungsgemeinschaft (DFG) under the Emmy Noether Programme (524408221 to CP; https://gepris.dfg.de/gepris/person/518862085?language=en) and the German Scholars Organisation through the Klaus Tschira Boost Fund (KT40 to CP; https://gsonet.org/foerderprogramme/klaus-tschira-boost-fund/). The funders had no role in study design, data collection and analysis, decision to publish, or preparation of the manuscript.

**Competing interests:** The authors have declared that no competing interests exist.

**Abbreviations:** dHCP, Developing Human Connectome Project; FDR, false discovery rate; LBO, Laplace–Beltrami operator; MRI, magnetic resonance imaging; SENSE, sensitivity encod-ing; TE, echo time; TR, repetition time; TSE, Turbo Spin Echo.

key steps in perinatal neurodevelopment [1–3], it remains unclear how these pro-cesses produce the regional heterogeneity of the entire human cortex.

Early cortical maturation is characterized by heterochronicity, in which cortical regions develop at different paces. For example, synaptic density in the primary visual cortex peaks during the 5th month of development, whereas in the prefrontal cortex, the peak occurs during the 15th postnatal month [4]. Regional differences in cortical phenotypes are initially driven by morphogens (e.g., FGF, Shh, and WNT), whose overlapping gradients result in a protomap of arealisation [5]. Neural activity progressively refines this protomap, shifting from predominately spontaneous syn-chronized bursts during gestation to more sensory-driven thalamocrortical activity postnatally [2]. While in utero the fetus is sensitive to the maternal heartbeat, touch of the abdomen and external noise [6,7], genetic programs primarily steer neurode-velopmental processes [8,9]. After birth, alongside internally generated activity (e.g., during sleep), neurodevelopment is increasingly shaped by experience-dependent plasticity, involving reinforcement of frequently used axons through myelination and pruning of infrequently used synapses [4,10,11]. These differences in intra- versus extrauterine development motivate consideration of how pre- and postnatal durations impact the architecture of the infantile cortex. In addition, understanding the impact of each developmental period will allow us to gain insight into how premature birth disrupts neurodevelopmental processes, and increases the risk of neurodevelopmen-tal disorders [12,13].

Magnetic resonance imaging (MRI) studies have shown regionally variable pat-terns of cortical maturation [14–16]. Across multiple modalities, differences in cortical development are most prominent between sensory and association cortices, suggest-ing that unimodal areas develop earlier than heteromodal areas [17–19]. However, extant in vivo research has primarily been constrained to macroscale features, such as thickness and surface area, which are relatively insensitive to finer-scale features, e.g., cortical columns and layers [20,21]. Yet these finer-scale features are progres-sively refined during perinatal neurodevelopment and lay the foundation for functional specialization of each cortical region [1,22], therefore there is a need to shift the focus on investigating microstructure rather than morphology alone.

Intracortical profiling of sub-millimeter resolution in vivo MRI offers a novel oppor-tunity to tap into the micro-scale maturation of the cortex. Recent work has shown that microstructure profiles, extracted from T1-weighted/T2-weighted (T1w/T2w) imaging, capture regional differences in laminar organization, as indicated by post-mortem histology [23–25]. T1w/T2w imaging enhances cortical myelin contrast by mathematically canceling out receive field (B1−) bias [26–28]. The residual transmit field (B1+) biases, however, vary across the field of view and across participants in such a way that absolute differences in T1w/T2w values cannot be directly inter-preted as absolute differences in cortical myelin [29,30]. Inspecting relative T1w/T2w levels across cortical depths within an area overcomes this issue, however, due to the low spatial frequency of the B1+ biases [31] Indeed, using a multi-modal test-retest dataset [32], Paquola and colleagues [23] recently showed that the shape of T1w/T2w intracortical profiles are reliable and exhibit very similar individual-specific

spatial patterns to those derived from quantitative R1. Combined with evidence that R1 is strongly correlated with histological myelin [33], these findings support that the intracortical balance of T1w/T2w effectively proxies depth-wise variations in myelin, and these values can be compared across participants.

In this study, we leveraged high-resolution MRI from 599 infants in their first postnatal weeks, spanning a wide spectrum of gestational ages (23–42 weeks), to characterize the maturational dynamics of cortical microstructure. We evaluated how intra- and extrauterine durations impact cortical development and its progression, as well as how modeling gestational and postnatal ages separately can help explain the regional differences in the influence of dominantly intrinsic, prenatal programming from increasingly experience-dependent postnatal changes on cortical microstructure. Finally, we assessed to what extent these dynamics are constrained by low-dimensional cortical geometry and thereby reflect patterning by spatial gradients.

## Methods

### Participants and demographics

The present study leveraged the Developing Human Connectome Project (dHCP), an open-access initiative designed to map neonatal brain development using multi-modal MRI (https://www.developingconnectome.org/, [34]). The study was approved by the United Kingdom Health Research Authority (Research Ethics Committee reference number: 14/LO/1169) and conducted in accordance to the principles expressed in the Declaration of Helsinki. Written parental consent was obtained in every case for imaging and open data release of the anonymized data. In the 4th release of the dataset (https://biomedia.github.io/dHCP-release-notes/), 805 neonate participants (373 females, 26–45 weeks postmenstrual age) with a total of 915 scans were provided, including term-born ($n = 594$) and prematurely born ($n = 211$) infants. Infants born at 37 weeks of gestational age or earlier are classified as preterm [35]. Exclusion criteria were chromosomal abnormalities, brain injuries, contraindications for MRI scanning, and acute birth complications treated with prolonged resuscitation. Among the 805 participants, 100 originated from non-singleton pregnancies. Additionally, a subsample of neonates were scanned more than once, with the 4th release including two timepoints for 106 participants and three timepoints for two participants.

Here, we examined the influence of three different developmental measures: gestational age, postnatal age, and postmenstrual age. Gestational age captures the intrauterine development of the fetus and is calculated as the duration from the first day of the mother's last menstrual period to the day of birth. Postnatal age refers to the period from birth to the day of the scan, reflecting the extrauterine development of the infant. Finally, postmenstrual age is the sum of the former two metrics, representing the cumulative maturation of the brain during both the pre- and postnatal stages of development. Since gestational and postnatal age are not independent of postmenstrual age, it is not possible to biologically separate the influences of each period. However, we aimed to partially disentangle their distinct associations with cortical microstructure conditional on postmenstrual age, in lieu of being able to reconstruct individualized developmental trajectories throughout the *in- and ex-utero* periods. All ages/developmental measures are reported in weeks$^{+days}$, e.g., "6$^{+6}$ weeks".

For our study, additional inclusion criteria were the availability of T1w and T2w images, as well as a complete FreeSurfer output. Furthermore, for participants with longitudinal scanning, we only used scans from the earliest time point to maintain a balanced cross-sectional dataset and avoid biases introduced by repeated measures. Finally, since the oldest term-born infant had a postnatal age of 6$^{+6}$ weeks, scans of preterm neonates with a postnatal age exceeding 7 weeks were also excluded to cover the mismatch in postnatal age distributions. This also worked as an exclusion criterion for preterm participants which were scanned significantly later after birth due to medical complications. The final dataset consisted of 599 participants (279 females, 162 preterm, 79 from non-singleton pregnancies). The inclusion of preterm infants in the study aims at providing a wide range of both pre- and postnatal developmental periods to be able to estimate the independent association of gestational age with intracortical microstructure. Results were replicated using the same

PLOS Biology

sample of participants but excluding the participants of non-singleton pregnancies (S1–S3 Figs), as well as excluding extremely and very preterm infants (S4–S6 Figs). Full demographics of the subsample used in this study are presented in Table 1 and Fig 1.

## MRI acquisition and preprocessing

The MRI acquisition was performed at St. Thomas Hospital, London using a Philips 3T scanner. The acquisition protocol was optimized for neonatal imaging, incorporating custom-built head coils as well as specialized positioning and immobilization techniques to minimize motion artifacts, while ensuring the safety of the participants [34]. Participants were naturally asleep during the image acquisition with the exception of six out of the 805 subjects, for whom sedation was used (two of which included in the present study). T1 images were acquired using an IR (Inversion Recovery) TSE sequence at a resolution of 0.8 × 0.8 × 1.6 mm with a repetition time (TR) of 4.8 s, an echo time (TE) of 8.7 ms, and sensitivity encoding (SENSE) factors of 2.26 (axial) and 2.66 (sagittal). T2 images were acquired at the same resolution using a Turbo Spin Echo (TSE) sequence, with a TR of 12 s, TE of 156 ms, and SENSE factors of 2.11 (axial) and 2.58 (sagittal).

Preprocessed data were downloaded from the 4th release of the dHCP. For comprehensive details on image preprocessing, refer to Makropoulos and colleagues [36]. In brief, motion correction and super-resolution reconstruction were employed on T1 and T2 images to achieve isotropic volumes with a resolution of 0.5 mm³ [37,38]. Then, brain tissue was

**Table 1. Dataset demographics and developmental metrics.**

| | Median ± SD (range) | Sex differences | Bivariate correlation with PMA (r) | Bivariate correlation with GA (r) | Bivariate correlation with PNA (r) |
|---|---|---|---|---|---|
| **Postmenstrual age (PMA, weeks)** | 40$^{+3}$ (26$^{+5}$–44$^{+5}$) | $t(597) = 0.29$ $p = 0.77$ | – | 0.92 | −0.04 |
| **Gestational age (GA, weeks)** | 39$^{+2}$ (23$^{+5}$–42$^{+2}$) | $t(597) = 0.10$ $p = 0.92$ | 0.92 | – | −0.43 |
| **Postnatal age (PNA, weeks)** | 0$^{+6}$ (0.00–7.00) | $t(597) = 0.42$ $p = 0.68$ | −0.04 | −0.43 | – |
| **Weight at scan (kg)** | 3.06 ± 0.92 (0.74–5.36) | $t(559) = -2.25$ $p = 0.03$ | 0.88 | 0.83 | −0.10 |
| **Head circumference at scan (cm)** | 33.67 ± 3.14 (21.00–39.10) | $t(571) = -2.44$ $p = 0.02$ | 0.88 | 0.83 | −0.08 |

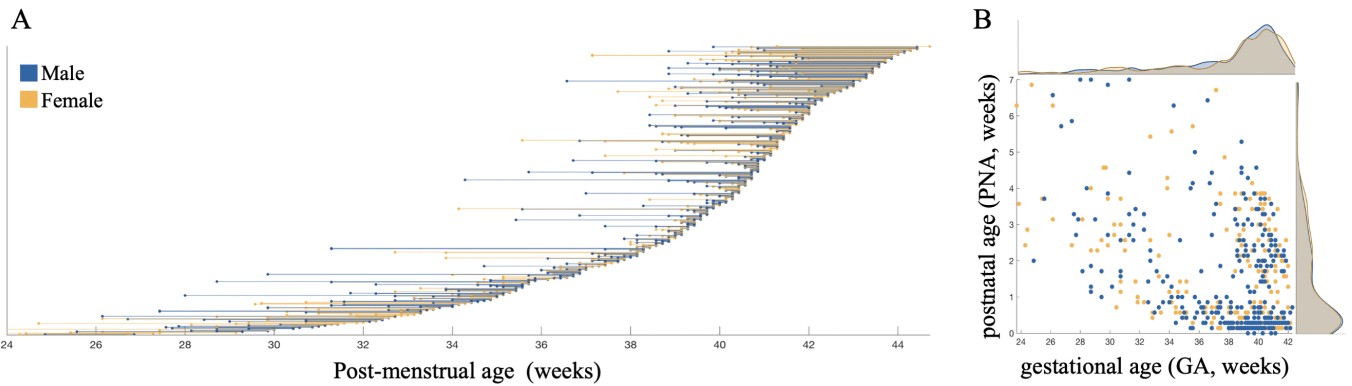

**Fig 1. Dataset demographics. (A)** Age distributions of the 599 participants (y-axis). The plot represents duration of gestation at time of birth (left dot) and age at time of scan (right dot). **(B)** The scatter plot illustrates the relationship between gestational age (GA) and postnatal age (PNA), with data points color-coded by sex (blue: male, yellow: female). Histograms on the edges of the scatter plot show the distributions of gestational (top) and postnatal (right) age for males and females.

extracted from the T2 images (BET, [39]), tissue-type segmentation was performed using Draw-EM [40] and cortical surfaces were reconstructed from T1 images. Following rigid co-registration of the motion-corrected T1w to the T2w image, T1w/T2w ratio images were computed.

To minimize potential inaccuracies of vertex-to-vertex correspondence, we parcellated cortical measurements into broader regions using the von Economo atlas, which captures regional differences in laminar organization [41,42]. We aligned the parcellation scheme to native cortical surfaces via the dhcpSym40 40-week (postmenstrual age) infant template surface [43]. For this purpose, we used template-to-native registration spheres that were released with dHCP and multimodal surface matching [44], optimized for alignment of sulcal depth. Regions within the limbic lobe (von Economo areas $L_{A1}$, $L_{A2}$, $L_{C1}$, $L_{C2}$, $L_{C3}$, $L_D$, and $L_E$) were excluded from the analysis due to limited cortical thickness in these regions, which in turn reduced the capacity of intracortical intensity sampling. This omission resulted to 66 parcels (33 in each hemisphere). For validation of the results, all analyses were replicated using the Schaefer-200 atlas [45] (S7 and S8 Figs).

## Microstructure profiles and their central moments

To analyze the cortical microstructure of the participants, we constructed 12 equivolumetric surfaces between native pial and white matter surfaces. The equivolumetric model compensates for variations in laminar thickness between sulci and gyri [46,47] by varying the Euclidean distance between pairs of intracortical surfaces based on curvature to preserve the fractional volume between surfaces [48]. Then, we sampled image intensities at matched vertices across the 12 intracortical depths using trilinear interpolation, which calculates a weighted average of eight neighboring voxels at each sampling depth (Fig 2A). The average spatial distributions of T1w/T2w intensities can be seen in S14 Fig.

To achieve a low-dimensional representation of the complex patterns of intracortical variation, we characterized the shape of each profile using two central moments. This method, inspired by histological studies [49], provides a compact, biological interpretable description of the signal distributions. Computation of $\mu 1$ and $\mu 2$ involves treating the profile as a histogram with intracortical depths as bins and the intensities as frequency, then simply computing the mean and standard deviation values. Thereby, the first moment, *center of gravity ($\mu_1$)*, describes how the intensity distribution is balanced across cortical depths, whereby a high center of gravity corresponds to higher intensities in the deeper cortex and a low center of gravity describes higher intensities in the superficial cortex, hence providing an indicator of the balance of microstructure across supra- and infragranular layers (Fig 2Bi). The second moment, *variance ($\mu_2$)*, characterizes the spread of intensities across the profile. A higher variance relates to a flatter intensity profile with a more homogeneous intensity distribution along depths (i.e., uniformly distributed intensities across intracortical depths), and a low variance describes a heterogeneous intensity distribution across intracortical depths (Fig 2Bii). Each of these moments, as well as their combinations, provides insights into the nature of lamination in the cortex and captures distinct patterns of microstructural differentiation across the cortical surface (Fig 2C). A recent study has tested the validity of using central moments in lower-resolution data by downsampling ultra-high-resolution T1 data [23,50]. The central moment maps of the downsampled data were highly similar to the maps of the original data ($r > 0.9$), a finding supporting that, even at the resolution of neonatal data, microstructure profiling can capture meaningful, regional variations in laminar organization.

Partial volume effects with white matter are unavoidable at the resolution of the analyzed scans, however, previous work has shown that correcting for these effects has little to no impact on the central moments [51], since these describe the intracortical distribution of intensities instead of depth-specific intensities. Additionally, the boundary between the gray and white matter is not sharply defined, therefore signal-mixing in deeper layers may index interesting developmental changes in long-range projections and the underlying U-fiber system. Together, the equivolumetric spacing, the oversampling using trilinear interpolation, as well as the use of central moments to summarize the intracortical profiles ensure a robust and thickness-insensitive characterization of intracortical T1w/T2w distributions at the resolution available. This approach enables the reliable investigation of the regional variation of laminar organization, somewhat overcoming the limitation of the relatively low resolution of the scans.

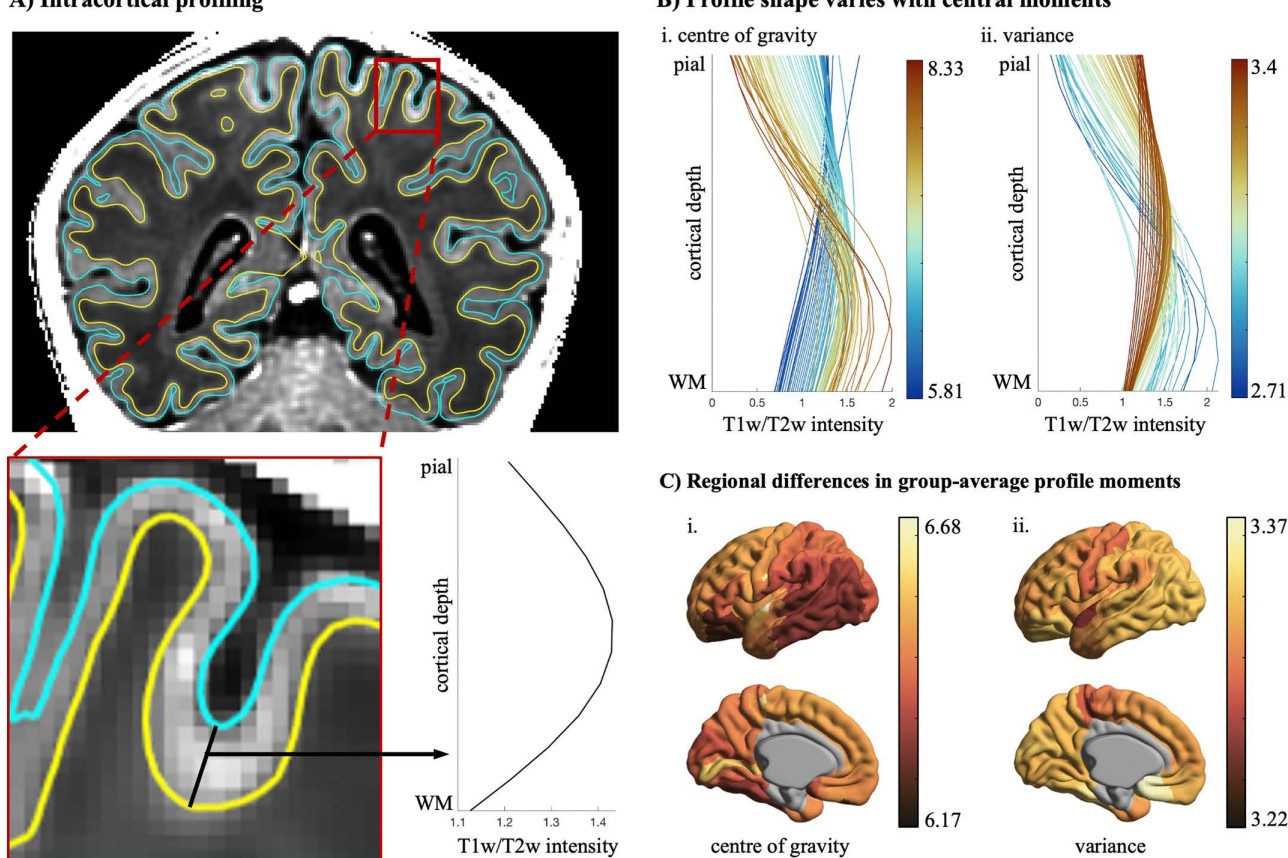

**Fig 2. Intracortical profiling on infant MRI. (A)** Intracortical intensity profiles were extracted at each vertex of infant structural MRI scans. Intensities were sampled at 12 equivolumetric intracortical surfaces, spanning from the pial boundary (blue) to the white matter boundary (yellow), capturing signal variations across cortical depths, defined as microstructure profiles. **(B)** Systematic changes in profile shape with respect to (i) center of gravity and (ii) variance. To illustrate this relationship, all profiles (across participants and regions) were sorted according to the respective moment, then averaged within 100 bins. (C) Parcel-wise central moment distributions mapped on the dHCP 40-week surface template (see S1 Data). Excluded regions (i.e., von Economo areas LA1, LA2, LC1, LC2, LC3, LD, and LE and the cortical wall) are shown in gray. Due to the high inter-hemispheric correlation of the central moments ($r=0.79$ for center of gravity and $r=0.93$ for variance), we show only results of the left hemisphere for simplicity.

## Statistical analysis

The statistical analysis focused on investigating the effects of postmenstrual age, gestational age, and postnatal age on microstructure, defined by the central moments of average intracortical intensity profiles on a parcel level. These three developmental metrics were analyzed both independently and collectively to account for different aspects of cortical development. First, to assess cumulative maturation of the brain, parcel-wise linear regression analysis was performed using postmenstrual age and sex as predictors and each central moment individually as response variables. False discovery rate (FDR) correction with an alpha of 0.025 was applied to account for multiple comparisons.

Next, to dissect the effect of postmenstrual age on cortical microstructure into its pre- and postnatal components, we first performed a linear model comparison, with each average central moment across all parcels as the response variable and combinations of gestational age, postnatal age, and sex as predictors, validating the fit of these variables for modeling cortical microstructure. Due to the broad overlap of the distributions of gestational and postnatal age, we isolated the unique contributions of each developmental variable to cortical microstructure in multi-variate regression models, that

included gestational age, postnatal age, and sex. Since gestational and postnatal age are not independent from postmenstrual age, we used the multi-variate models to assess the effects of these developmental metrics as relative associations and not to directly compare their magnitudes. Results of univariate models of gestational and postnatal age are shown in S9 Fig. To validate the independence of the results from cortical thickness maturation, all results were replicated using models which additionally included cortical thickness as a predictor (S10–S12 Figs).

To test whether regional variation in development concords with large-scale spatial gradients, we computed the product-moment correlations between the postmenstrual, gestational, and postnatal age effects and the 2nd–4th geometric eigenmodes [52]. Geometric eigenmodes were computed by applying the Laplace–Beltrami operator (LBO) to the surface mesh of a population-average infant neocortex ("dhcpSym" 40-week, [43]), using code provided by Pang and colleagues [52]. The LBO treats the cortical surface as a Riemannian manifold and computes eigenfunctions of the surface Laplacian, capturing geometric patterns from coarse to fine scales. A 1st eigenmode is spatially uniform by construction and was therefore not included in the analysis. Product-moment correlations were performed on unthresholded vertex-wise maps and spin permutation testing [53] was performed to assess the significance of the correlation with the eigenmodes, while accounting for spatial autocorrelation in each map (10,000 permutations, alpha level of 0.025). In addition, we assessed the additive effects of eigenmodes in explaining age effects through a series of uni- then multi-variate linear regressions.

## Results

### Changes in cortical microstructure with postmenstrual age

Globally, intracortical myelin-sensitive signal increases with postmenstrual age, in a manner that generally leads to a more balanced distribution of the T1w/T2w signal across cortical depths ($r_{variance} = 0.51$, $p < 0.001$; Fig 3A and 3B). These changes resulted from signal increases in upper layers for superior regions and increases in deeper layers in inferior regions, as illustrated by the bimodal pattern of age-related changes in center of gravity (Fig 3A and 3B). These analyses show how perinatal development of intracortical myelin is region- and depth-specific, whereby unique developmental trajectories are expressed across the cortex.

### Effects of gestational and postnatal age on cortical development

Next, we divided the developmental period into gestational age and postnatal age. To examine how the timing of birth and the developmental period after birth influence early cortical development, we linearly modeled the relationship of each central moment with gestational age and postnatal age, while controlling for the effect of the other (Fig 4). This approach doesn't aim at completely separating prenatal and postnatal influences, but rather to investigate the systematic differences in intracortical microstructure between infants who spent relatively more time in utero and infants who spent more time *ex utero*. Gestational age effects mirrored the postmenstrual age effects reported above, emphasizing the importance of the in utero period in shaping perinatal microstructure. During the postnatal period, only increases of the center of gravity in inferior regions remained statistically significant. Interestingly, we observed decreases in profile variance, corresponding to a less flat profile, thus suggesting that cortical layers become more differentiable in the first postnatal weeks, in contrast to the increasing homogeneity during prenatal development. These effects were relatively consistent with a different parcellation scheme ($r > 0.7$ for center of gravity and $r > 0.59$ for variance; S7–S9 Figs) and when excluding extremely preterm ($r > 0.98$; S4 Fig), very preterm ($r > 0.89$; S5 Fig) and preterm infants ($r > 0.70$; S6 Fig). Based on these findings, the distribution of myelin-sensitive signal across cortical layers shifts in distinct, region-specific ways in each developmental period.

### Alignment of developmental effects on myelin to geometric eigenmodes

Next, we assessed whether large-scale spatial gradients could account for regional differences in early microstructural development (Fig 5). Indeed, prenatally, changes in center of gravity were correlated with the 2nd eigenmode, reflecting

## A) Regional effects of postmenstrual age on intracortical myelin

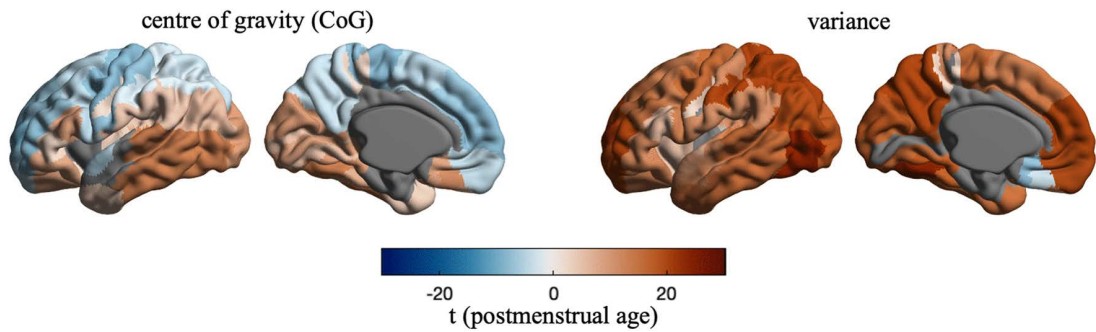

## B) Region-specific changes in intracortical profiles with postmenstrual age

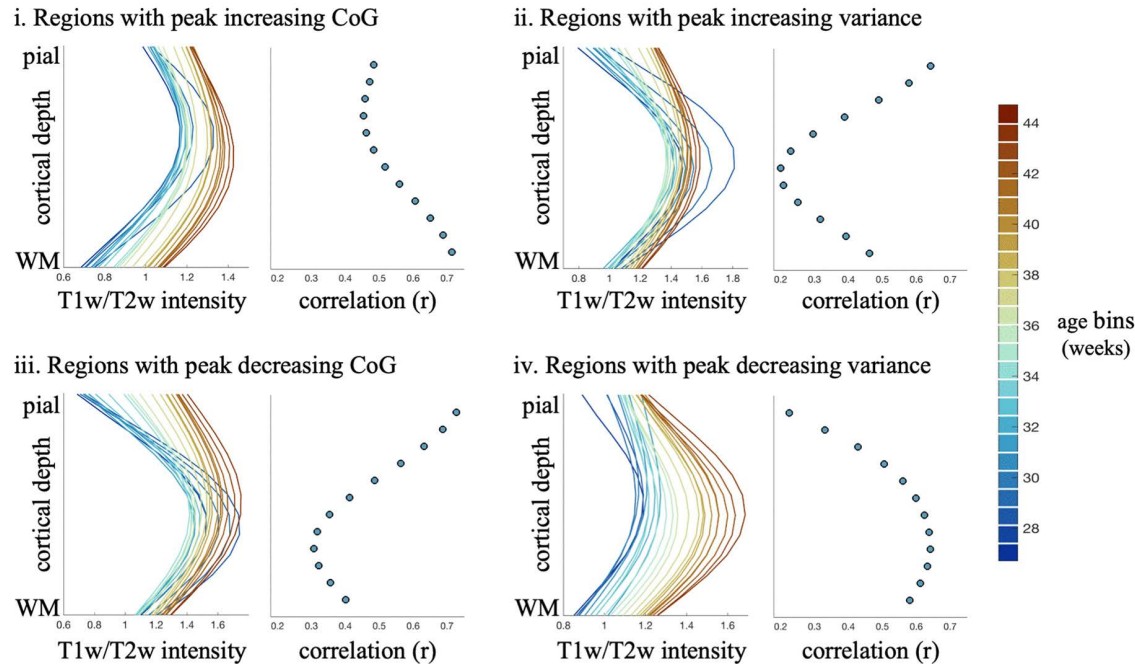

**Fig 3. Postmenstrual age and intracortical myelin. (A)** Linear regression models were used to assess the association between postmenstrual age (PMA) and intracortical profile moments across cortical regions, while controlling for sex. Surface maps display *t*-values for the PMA-estimate, projected onto the cortical surface for center of gravity (left) and variance (right) (see S1 Data). Statistically non-significant parcels (*p* > 0.05) and excluded parcels are displayed in gray. **(B)** Sliding window visualization of profile changes with PMA (age bins = 1.5-week windows with 50% overlap) for the two regions with the maximum increases and decreases for center of gravity (panel i and iii) and variance (panel ii and iv), with neighboring scatter plots depicting the relationship between PMA with depth-specific T1w/T2w-intensities based on product-moment correlations. Due to the high inter-hemispheric correlation of the PMA-effects (*r* = 0.84 for center of gravity and *r* = 0.96 for variance), we show only results of the left hemisphere for simplicity. Abbreviations: CoG, center of gravity; PMA, postmenstrual age.

an anterior–posterior gradient (*r* = −0.62, $p_{spin}$ = 0.03). Regional differences in postnatal age effects were also correlated with spatial axes, with changes in the center of gravity also being aligned with the anterior–posterior axis (*r* = −0.68, $p_{spin}$ = 0.006) and changes in variance being aligned with the 3rd eigenmode, capturing a superior-inferior axis ($r_{var}$ = 0.57, $p_{spin}$ = 0.04). Combining eigenmodes in multi-variate models explained even more variance in the spatial patterns (Fig 5B). Age-related changes in center of gravity were particularly strongly associated with eigenmodes ($R^2$ > 0.6), suggesting that

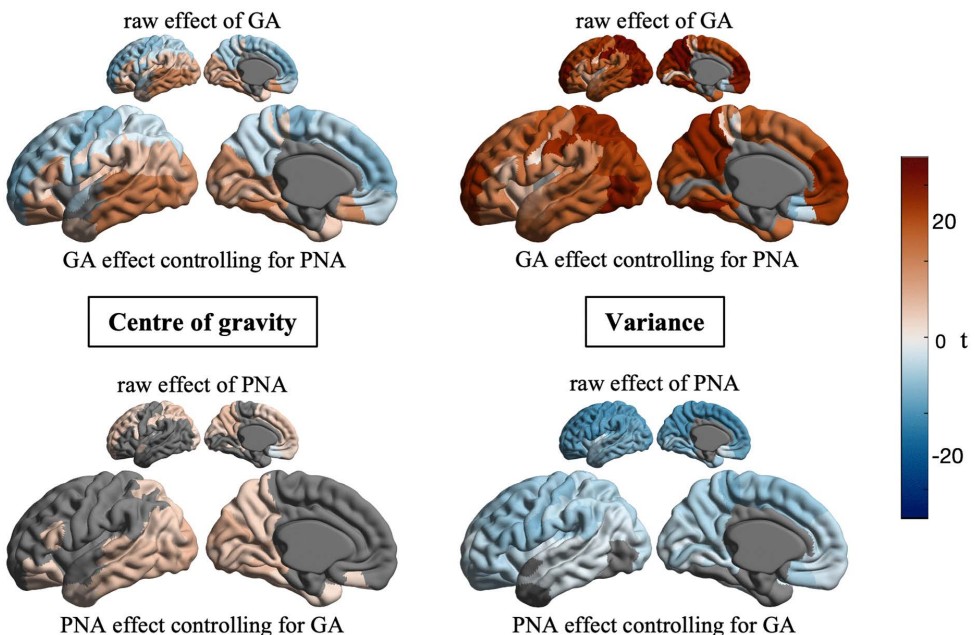

**Fig 4. Distinct associations of gestational and postnatal age with intracortical myelin.** Large cortical surface maps illustrate gestational (GA, top) and postnatal (PNA, bottom) age effects on each central moment, after controlling for PNA and GA, respectively. Smaller cortical surface maps illustrate univariate GA- and PNA-effects (i.e., without regressing out the other metric). Statistically non-significant parcels ($p > 0.05$) and excluded parcels are displayed in gray. Due to the high inter-hemispheric correlation of the GA- and PNA-effects (rGA = 0.84 and rPNA = 0.8 for center of gravity; rGA = 0.96 and rPNA = 0.9 for variance), we show only results of the left hemisphere for simplicity (see S1 Data).

the preference of microstructural increases to occur in upper versus lower layers is especially tightly linked to cortical geometry.

## Discussion

Late gestation and early postnatal life are pivotal windows for the development of the human cortex, during which changes in cortical cytoarchitecture set the foundations for the brain's functional specialization. In this study, we combined histology-inspired intracortical profiling [23,49] with T1w/T2w ratio [27] to characterize the developing structure of the infant human cortex. We identified regional heterogeneity in the structural changes that occur during the perinatal maturation of the cortex. Breaking down this perinatal period into its intra- and extrauterine stages revealed separable contributions of gestational and postnatal age to cortical development, highlighting how the structure of the cortex is differentially influenced by the internal environment during gestation and the external environment after birth. We also identified significant correlations between the developmental dynamics of myelin and geometric eigenmodes of the cortex [52], suggesting that early cortical maturation is influenced by topographical constraints.

The influences on cortical development shift at the time of birth. During intrauterine development intrinsic genetic programming primarily dictates the organization of cortical laminae, and subplate activity guides neuronal migration and axonal extension [54,55]. While sensory experience also shapes the cortical structure during gestation [56], after birth, the variety of sensory input and the maturation of GABAergic inhibition [57,58] shift the modulation of structural dynamics toward experience-dependent synaptogenesis and intracortical myelination [4,59]. In essence, perinatal development is characterized by a transition from a more strictly-regulated, genetically-governed maturation to increasingly experience-dependent changes in connection strength. This modulatory transition highlights the importance of the timing

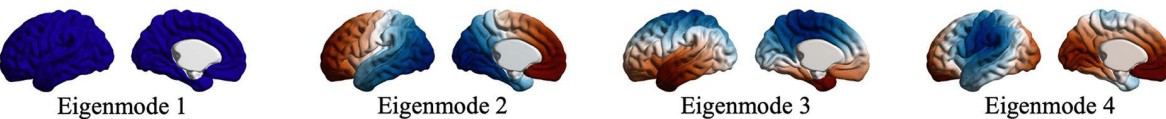

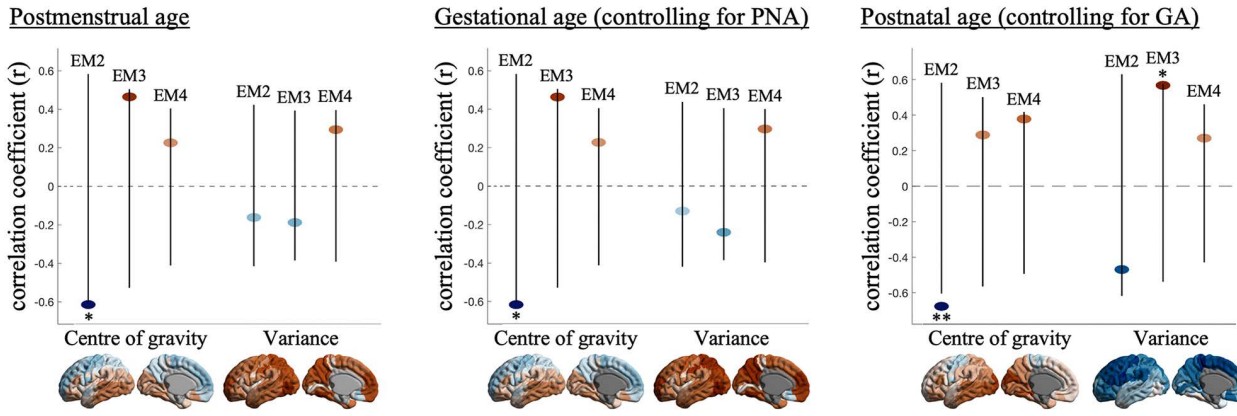

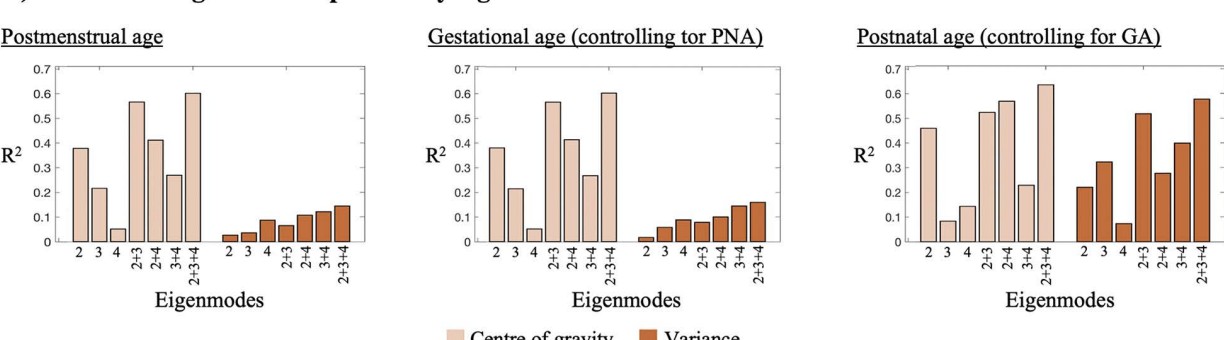

**Fig 5. Alignment of intracortical myelin development with geometric eigenmodes. (A)** Surface maps of the first four vertex-wise geometric eigenmodes (see S1 Data). **(B)** Product-moment correlation coefficients between the effects of postmenstrual (left), gestational (middle), and postnatal age (right) on each central moment and the 2nd–4th eigenmodes (see S1 Data). The surface maps shown below each scatter plot are the same maps shown in Figs 3 and 4, but without significance threshold. The whiskers represent the range of correlation values obtained from spin-based permutation testing ($n = 10,000$), between the 2.5th to 97.5th percentile of the permuted correlations. Significant correlations are highlight with one asterisk for $p < 0.05$ and two asterisks for $p < 0.01$. **(C)** Linear regression models were used to assess the explained variance of the effects of postmenstrual (left), gestational (middle), and postnatal age (right) by the 2nd–4th geometric eigenmodes or combinations thereof. Data and code used to generate this figure can be found at https://doi.org/10.5281/zenodo.18910237.

of birth in brain development, as prematurely born infants are exposed to extra-uterine sensory input at a stage when their cortical circuitry is still undergoing rapid organization, which may contribute to altered trajectories of postnatal plasticity. However, other significant clinical factors, such as mechanical ventilation, antenatal steroids, and oxygenation control, are contributing to the disrupted cortical growth of prematurely born infants.

Addressing the transitional period around birth, our study highlighted the importance of investigating and characterizing early cortical development in a way that assesses both the cumulative period of development (intra- and extrauterine),

as well as each of its components individually. In particular, we included preterm infants in our analyses to gauge a broad range of gestational durations and statistically compensated for the generally higher postnatal ages in preterm infants using multi-variate regression models. By modeling gestational and postnatal ages separately, we showed that the timing of birth dominated the effects on early microstructural development, highlighting the importance of the intrinsic programming. We also observed significant postnatal changes in several cortical regions, even after controlling for these prominent effects of gestational maturation and despite the short postnatal period investigated in this study. In occipital, posterior temporal, and some prefrontal regions, myelin increases are more prominent in the deeper cortical layers, following a similar trajectory to the gestational dynamics. In contrast, we did not observe a continuation of prenatal increases in upper cortical layers in frontal and central areas in the postnatal period, showing a potential discontinuity between developmental trajectories of the two periods. Finally, whereas intracortical profiles become increasingly balanced with later gestational age, this effect seemed to flip after birth with postnatal changes in laminar organization. In general, the prominent effect of gestational maturation on cortical myelin hints toward the importance of the timing of birth and is consistent with literature across various modalities supporting that premature transition to the *ex utero* environment may alter the otherwise well-regulated developmental trajectories of cortical growth [60–63]. However, when we explicitly tested the interaction term between gestational and postnatal age, we found significant effects in only about one quarter of the parcels when modeling center of gravity and none of the parcels when modeling variance. Thus, the idea that earlier transition to the *ex utero* environment influences the trajectories of postnatal cortical development should be viewed as a speculative interpretation, rather than a conclusion from our analyses. Longitudinal studies extending into later infancy would be especially helpful for elucidating how and when the modest postnatal age influences observed here evolve into larger, experience-driven changes. Such work could complement cross-sectional approaches by providing within-subject developmental trajectories.

Regional differences in cortical cytoarchitecture originate from morphogen gradients, which pattern the mammalian cortex with variable concentrations of growth factors during gestation [64]. Supporting the importance of gradients in shaping cortical development, we also found that large-scale eigenmodes capture a majority of variance in regional heterogeneity, especially in terms of changes in the balance between upper and lower cortical layers. Starting in utero and continuing during postnatal development, thalamocortical connectivity is thought to refine cortical arealisation, especially around primary sensory areas [54,65]. Relatedly, histological and imaging studies consistently show that primary sensory and motor regions mature earlier than the association cortex [15,16,66]. While the short postnatal timeframe encompassed by this dataset precludes interpretation of maturity per se, we observed strong postnatal changes in deeper layers of the occipital and inferior temporal cortex, highlighting a preferential development of deeper layers of cortical regions involved in visual processing during this early period of immense sensory stimulation. Notably, we found an opposing trend in sensorimotor areas, with a preference towards microstructural development of upper layers. This contrast may underpin the differentiation of sensorimotor and visual cortical areas; a pattern that aligns with their strong functional differentiation during infancy [67]. However, although intracortical profiling allows the characterization of depth-dependent differences between regions, central moment metrics are agnostic to the timing of neuronal migration and layer formation, therefore it's not possible to determine whether one of these regions is "more advanced" than the other in a developmental sense.

This study was constrained by a number of limitations. First, the inclusion of extremely and very preterm infants makes it difficult to separate the influences of gestational timing from the impact of medical interventions that concern prematurely born infants (e.g., respiratory support, medication exposure, etc.). Although we performed sensitivity analyses excluding extremely and very preterm infants, residual confounding by these clinical factors cannot be excluded. Therefore, our assumptions regarding premature transition to the *ex utero* environment altering developmental trajectories of cortical growth must be interpreted as hypothetical interpretations and not as definitive conclusions. Second, the cross-sectional design and narrow postnatal age range limit the potential to investigate developmental trajectories or to disentangle prenatal from postnatal influences. Gestational and postnatal age are collinear components of postmenstrual

age and differ in their ranges, thus we can only infer relative associations with these developmental components, rather than strict separations of pre- versus postnatal influences. Finally, although our approach provides novel insights into the balance of myelin across cortical depths, the imaging resolution prohibits the delineation of separate cortical layers. As such, we may not make conclusions about the differentiation or development of numbered layers.

In conclusion, the present study demonstrates how intracortical profiling of neonatal MRI can reveal distinct spatial patterns of development, which reflect the regional differentiation across large-scale spatial gradients. These findings offer a framework for linking macroscale in-vivo MRI to microscale biology, showing how depth-specific changes underpin regional differentiation between sensory systems and levels of the cortical hierarchy. Modeling gestational and postnatal ages separately helped to disentangle intrinsic, prenatal influences from more experience-dependent postnatal changes of cortical microstructure, illustrating how intrauterine development dominates microstructural refinement in comparison to the very early stages of extrauterine development. Our approach lays the groundwork for future studies to elucidate the biological underpinnings that govern normative cortical maturation, as well as to pinpoint how prematurity increases the risk of neurodevelopmental disorders [68–70].

## Supporting information

**S1 Fig. Effects of postmenstrual age on cortical myelin, excluding twin participants.** Linear regression models were used to assess the association between postmenstrual age (PMA) and intracortical profile moments across cortical regions, while controlling for sex. Bigger surface maps were derived from a subsample of participants, which excluded any participants of multiple pregnancies. Smaller surface maps were derived from the original sample of this study. All surface maps display *t*-values for the PMA-estimate, projected onto the cortical surface for center of gravity (top) and variance (bottom). The spatial correlation between the effects on the original dataset and the dataset excluding twins is displayed in the middle of each set, together with the *p*-value derived from spin-based permutation testing (n = 10,000), between the 2.5th to 97.5th percentile of the permuted correlations. Excluded parcels are displayed in gray.
(PDF)

**S2 Fig. Effects of gestational age on cortical myelin after correcting for the effects of postnatal age, excluding twin participants.** Linear regression models were used to assess the association between gestational age (GA) and intracortical profile moments across cortical regions, while controlling for sex. Bigger surface maps were derived from a subsample of participants, which excluded any participants of multiple pregnancies. Smaller surface maps were derived from the original sample of this study. All surface maps display *t*-values for the GA-estimate, projected onto the cortical surface for center of gravity (top) and variance (bottom). The spatial correlation between the effects on the original dataset and the dataset excluding twins is displayed in the middle of each set, together with the *p*-value derived from spin-based permutation testing (n = 10,000), between the 2.5th to 97.5th percentile of the permuted correlations. Excluded parcels are displayed in gray.
(PDF)

**S3 Fig. Effects of postnatal age on cortical myelin after correcting for the effects of gestational age, excluding twin participants.** Linear regression models were used to assess the association between postnatal age (PNA) and intracortical profile moments across cortical regions, while controlling for sex. Bigger surface maps were derived from a subsample of participants, which excluded any participants of multiple pregnancies. Smaller surface maps were derived from the original sample of this study. All surface maps display *t*-values for the PNA-estimate, projected onto the cortical surface for center of gravity (top) and variance (bottom). The spatial correlation between the effects on the original dataset and the dataset excluding twins is displayed in the middle of each set, together with the *p*-value derived from spin-based permutation testing (n = 10,000), between the 2.5th to 97.5th percentile of the permuted correlations. Excluded parcels are displayed in gray.
(PDF)

**S4 Fig. Effects on cortical myelin, excluding extremely preterm infants.** Linear regression models were used to asses the association between postmenstrual (left), gestational (middle) and postnatal age (right) and intracortical profile moments across cortical regions, while excluding extremely preterm (EPT) infants from the analyses. The first and third rows of surface maps display the *t*-values for the developmental variables in the original dataset with all participants, whereas the second and forth rows display the *t*-values for the developmental variables in the subset excluding EPT participants. The first two rows correspond to the linear models with center of gravity as response variable, whereas the third and forth rows correspond to the models with variance as response variable. Excluded parcels are displayed in gray. (PDF)

**S5 Fig. Effects on cortical myelin, excluding very preterm infants.** Linear regression models were used to asses the association between postmenstrual (left), gestational (middle) and postnatal age (right) and intracortical profile moments across cortical regions, while excluding very preterm (VPT) infants from the analyses. The first and third rows of surface maps display the *t*-values for the developmental variables in the original dataset with all participants, whereas the second and forth rows display the *t*-values for the developmental variables in the subset excluding VPT participants. The first two rows correspond to the linear models with center of gravity as response variable, whereas the third and forth rows correspond to the models with variance as response variable. Excluded parcels are displayed in gray. (PDF)

**S6 Fig. Effects on cortical myelin, excluding preterm infants.** Linear regression models were used to asses the association between postmenstrual (left), gestational (middle) and postnatal age (right) and intracortical profile moments across cortical regions, while excluding preterm (PT) infants from the analyses. The first and third rows of surface maps display the *t*-values for the developmental variables in the original dataset with all participants, whereas the second and forth rows display the *t*-values for the developmental variables in the subset excluding PT participants. The first two rows correspond to the linear models with center of gravity as response variable, whereas the third and forth rows correspond to the models with variance as response variable. Excluded parcels are displayed in gray. (PDF)

**S7 Fig. Effects of postmenstrual age on cortical myelin, Schaefer-200 atlas.** Linear regression models were used to assess the association between postmenstrual age (PMA) and intracortical profile moments across cortical regions of the Schaefer-200 atlas, while controlling for sex. Bigger surface maps display *t*-values for the PMA-estimate, projected onto the cortical surface for center of gravity (top) and variance (bottom). Smaller surface maps show the results of the same models using the von Economo atlas. The spatial correlation between the effects on the Schaefer-200 and von Economo atlas is displayed in the middle of each set, together with the *p*-value derived from spin-based permutation testing ($n = 10,000$), between the 2.5th to 97.5th percentile of the permuted correlations. To calculate these correlations, parcel-wise results were upsampled to vertex-wise level. Excluded parcels are displayed in gray. (PDF)

**S8 Fig. Effects of gestational age on cortical myelin, after correcting for the effects of postnatal age, Schaefer-200 atlas.** Linear regression models were used to assess the association between gestational age (GA) with intracortical profile moments across cortical regions of the Schaefer-200 atlas, while controlling for sex. Bigger surface maps display *t*-values for the GA-estimate, projected onto the cortical surface for center of gravity (top) and variance (bottom). Smaller surface maps show the results of the same models using the von Economo atlas. The spatial correlation between the effects on the Schaefer-200 and von Economo atlas is displayed in the middle of each set, together with the *p*-value derived from spin-based permutation testing ($n = 10,000$), between the 2.5th to 97.5th percentile of the permuted correlations. To calculate these correlations, parcel-wise results were upsampled to vertex-wise level. Excluded parcels are displayed in gray. (PDF)

**S9 Fig. Effects of postnatal age on cortical myelin, after correcting for the effects of gestational age, Schaefer-200 atlas.** Linear regression models were used to assess the association between postnatal age (PNA) with intracortical profile moments across cortical regions of the Schaefer-200 atlas, while controlling for sex. Bigger surface maps display *t*-values for the PNA-estimate, projected onto the cortical surface for center of gravity (top) and variance (bottom). Smaller surface maps show the results of the same models using the von Economo atlas. The spatial correlation between the effects on the Schaefer-200 and von Economo atlas is displayed in the middle of each set, together with the *p*-value derived from spin-based permutation testing ($n = 10,000$), between the 2.5th to 97.5th percentile of the permuted correlations. To calculate these correlations, parcel-wise results were upsampled to vertex-wise level. Excluded parcels are displayed in gray. (PDF)

**S10 Fig. Depth-wise intensity distributions.** Parcel-wise T1w/T2w intensity distributions mapped on the dHCP 40-week surface template. Excluded regions (i.e., von Economo areas $L_{A1}$, $L_{A2}$, $L_{C1}$, $L_{C2}$, $L_{C3}$, $L_D$, and $L_E$ and the cortical wall) are shown in gray. (PDF)

**S11 Fig. Effects of gestational and postnatal age on cortical myelin, without correcting for the effects of each other.** Cortical surface maps illustrate the spatial distribution of gestational (left) and postnatal (right) age effects on each central moment, without controlling for postnatal and gestational age, respectively. The parcel-wise estimates were obtained from linear regression models with gestational age and sex or postnatal age and sex as predictors. Excluded parcels are displayed in gray. (PDF)

**S12 Fig. Effects of postmenstrual age on cortical myelin, after correcting for the effects of cortical thickness.** Linear regression models were used to assess the association between postmenstrual age (PMA) and intracortical profile moments across cortical regions, while controlling for cortical thickness and sex. The original results of this study, without controlling for cortical thickness are displayed as smaller surface maps. All surface maps display *t*-values for the PMA-estimate, projected onto the cortical surface for center of gravity (top) and variance (bottom). The spatial correlation between the effects with and without controlling for cortical thickness is displayed in the middle of each set, together with the *p*-value derived from spin-based permutation testing ($n = 10,000$), between the 2.5th to 97.5th percentile of the permuted correlations. Excluded parcels are displayed in gray. (PDF)

**S13 Fig. Effects of gestational age on cortical myelin, after correcting for the effects of cortical thickness.** Linear regression models were used to assess the association between gestational age (GA) and intracortical profile moments across cortical regions, while controlling for cortical thickness and sex. The original results of this study, without controlling for cortical thickness are displayed as smaller surface maps. All surface maps display *t*-values for the GA-estimate, projected onto the cortical surface for center of gravity (top) and variance (bottom). The spatial correlation between the effects with and without controlling for cortical thickness is displayed in the middle of each set, together with the *p*-value derived from spin-based permutation testing ($n = 10,000$), between the 2.5th to 97.5th percentile of the permuted correlations. Excluded parcels are displayed in gray. (PDF)

**S14 Fig. Effects of postnatal age on cortical myelin, after correcting for the effects of cortical thickness.** Linear regression models were used to assess the association between postnatal age (PNA) and intracortical profile moments across cortical regions, while controlling for cortical thickness and sex. The original results of this study, without controlling for cortical thickness are displayed as smaller surface maps. All surface maps display *t*-values for the PNA-estimate, projected onto the cortical surface for center of gravity (top) and variance (bottom). The spatial correlation between the

effects with and without controlling for cortical thickness is displayed in the middle of each set, together with the *p*-value derived from spin-based permutation testing (*n* = 10,000), between the 2.5th to 97.5th percentile of the permuted correlations. Excluded parcels are displayed in gray.
(PDF)

**S1 Data. Data file.**
(XLSX)

## Author contributions

**Conceptualization:** Casey Paquola.

**Data curation:** Stuart Oldham, Casey Paquola.

**Formal analysis:** Thanos Tsigaras.

**Funding acquisition:** Casey Paquola.

**Investigation:** Thanos Tsigaras, Juergen Dukart, Simon B. Eickhoff, Casey Paquola.

**Methodology:** Stuart Oldham, Casey Paquola.

**Project administration:** Casey Paquola.

**Resources:** Simon B. Eickhoff.

**Software:** Casey Paquola.

**Supervision:** Casey Paquola.

**Visualization:** Thanos Tsigaras, Casey Paquola.

**Writing – original draft:** Thanos Tsigaras, Casey Paquola.

**Writing – review & editing:** Thanos Tsigaras, Juergen Dukart, Stuart Oldham, Simon B. Eickhoff, Casey Paquola.

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
