## [Editor Report · Decision Letter 0]

20 Aug 2025

Dear Dr Paquola,

Thank you for submitting your manuscript entitled "Disentangling the influences of pre- and postnatal periods on human cortical microstructure" for consideration as a Research Article by PLOS Biology.

Your manuscript has now been evaluated by the PLOS Biology editorial staff and I am writing to let you know that we would like to send your submission out for external peer review.

Once your full submission is complete, your paper will undergo a series of checks in preparation for peer review. After your manuscript has passed the checks it will be sent out for review. To provide the metadata for your submission, please Login to Editorial Manager (https://www.editorialmanager.com/pbiology) within two working days, i.e. by Aug 22 2025 11:59PM.

Kind regards,

Christian

Christian Schnell, PhD

Senior Editor

PLOS Biology

cschnell@plos.org

---

## [Decision Letter · Decision Letter 1]

10 Nov 2025

Dear Dr Paquola,

Thank you for your patience while your manuscript "Disentangling the influences of pre- and postnatal periods on human cortical microstructure" was peer-reviewed at PLOS Biology. First of all, please allow me to apologize for the long delay in sending our decision. Unfortunately, it has been a bit more difficult than usual to find reviewers with the required expertise to review your manuscript. However, your manuscript has now been evaluated by the PLOS Biology editors, an Academic Editor with relevant expertise, and by several independent reviewers.

In light of the reviews, which you will find at the end of this email, we would like to invite you to revise the work to thoroughly address the reviewers' reports.

As you will see below, the reviewers expressed interest in your approach and find the study potentially valuable. However, all reviewers wrote that the text contains imprecise statements and needs greater nuance in its interpretations. Reviewer 1 and Reviewer 3 in particular also mention major technical concerns about the analytical approach, about aspects of the analyses and interpretations, and highlighted missing methodological information. We think that all these concerns need to be addressed. Moreover, in particular in light of the concerns about the spatial resolution, we ask you to address these concerns as far as possible by providing additional analyses and phrase the corresponding results and claims more carefully, and to discuss the limitations of your approach in a Limitations section within the Discussion.

Given the extent of revision needed, we cannot make a decision about publication until we have seen the revised manuscript and your response to the reviewers' comments. Your revised manuscript is likely to be sent for further evaluation by all or a subset of the reviewers.

**IMPORTANT - SUBMITTING YOUR REVISION**

*Re-submission Checklist*

*Published Peer Review*

*PLOS Data Policy*

*Blot and Gel Data Policy*

Sincerely,

Christian

Christian Schnell, PhD

Senior Editor

PLOS Biology

cschnell@plos.org

REVIEWS:

Reviewer #1: The manuscript presents a novel analysis of neonatal intracortical microstructure. Using a very large dataset of anatomical infant MRI, the authors sampled T1w/T2w ratios across equidistant surfaces spanning the cortical depth to model signal variation from pial to white matter. Profiles were summarized by mean intensity, centre of gravity, and variance, and developmental trajectories were examined with respect to gestational and postnatal age, then related to low spatial-frequency geometric eigenmodes. The approach of probing within-tissue properties and sublaminar features is interesting and potentially valuable. The study is generally well organized and clearly written. However, I have several concerns, especially regarding methodological limitations inherent to infant imaging and profile-based analyses, and some interpretations.

Major comments

1. T1w/T2w ratio interpretation and terminology

* Terms such as 'microstructural density' or 'increase in microstructure' are imprecise. Microstructure is not a direct metric; please describe the actual measured quantities (mean T1w/T2w values, variance etc). I'm not sure I understand what 'microstructural density' means in this context.

* Framing of T1w/T2w as a direct measure of microstructure should be interpreted with caution. Actually, you provide more details in Methods when describing the T1w/T2w, some of it repeats the introduction and some expands on it. Consider merging and using the Introduction to frame the concept as a composite marker of cortical microstructure from the start.

Methodological limitations

* Sampling across 12 equivolumetric intracortical surfaces is interesting, but neonatal cortex is very thin (<2 mm, 3-4 voxels at 0.5 mm resolution). Surfaces are thus highly non-independent and susceptible to partial volume effects, interpolation errors, and noise. Regional differences in cortical thickness may also affect centre of gravity and variance estimates because of 'closer/wider' sampling, potentially impacting the resulting 'laminar' intensity distributions. It would be good to discuss this more explicitly.

* Von Economo atlas is based on adult cytoarchitecture, raising concerns about the relevance of its regional boundaries in neonates, especially since your study focuses on laminar microstructure with goal to show that it significantly changes during the perinatal period (do you expect the laminae to change with age but the boundaries to stay exactly same as in adult?)

* Generally, methodological limitations are not discussed in the Discussion at all; including these would better contextualize the results.

* 'Histological studies, that do directly assess cytoarchitectural features, are naturally cross-sectional and often very limited in sample size.' Your study is also cross-sectional 0: )

Framing of perinatal development.

* The manuscript often presents perinatal development as a dichotomous shift from genetic to experience-driven processes. Ontogeny is continuous and shaped by complex interactions (maternal-fetal environment, epigenetics, metabolism, circuit maturation, sensory input). I recommend softening this framing and clarifying that observed changes reflect interacting processes rather than a strict intra- vs. extrauterine shift. I have also few hesitations about statements like:

o 'foetus is relatively isolated from sensory input' is not fully accurate. Fetal sensory input exists and differs from postnatal exposure.

o The claim that thalamocortical connectivity refines cortical arealisation postnatally is overly restrictive. Thalamocortical and corticocortical connections develop prenatally, with subplate activity already influencing laminar organization. Consider clarifying that these processes begin in utero and continue postnatally.

* While trying to distinguish prenatal vs. postnatal effects is interesting, I'm not sure the data allow strong conclusions. You have access to scans at a specific age (PMA), and then you create prenatal and postnatal period as summative age birth and birth-scan (24wGA +6 weeks to scan does not represent the same developmental period as 38wGA + 6 weeks and correcting with linear regression is probably insufficient to disentangle these differences).

* Also, GA and PNA differ in duration (months vs. max 7 weeks), so caution is warranted when interpreting relative effects (for example saying prenatal development has larger effects).

* Observed laminar differences (sensorimotor deeper vs. visual upper layers) are intriguing. You link this to later functional differentiation in discussion. Would this potentially reflect developmental asynchronies; sensorimotor areas might mature earlier, and superficial-to-deep maturation may place them in a 'deeper stage' at the observed ages (tentative).

Handling of preterm and non-singleton infants

* 'This modulatory transition highlights the importance of the timing of birth in brain development, with prematurely born infants being exposed to a sensory-rich environment before the scaffold of their circuitry has been completely established for postnatal plasticity.' Exactly, the premature babies should not be considered as developing 'typically' so mixing them into your population might be problematic. 30% of the sample is preterm, which complicates interpretation of GA vs. PNA effects. This might need to be explicitly acknowledged.

* The rationale for excluding scans with PNA >7 weeks is unclear. You assume equivalent postnatal weeks after preterm vs. term birth reflects the same developmental stage, it might be clarified bit more.

* 'In general, the prominent effect of gestational maturation on cortical microstructure emphasises the importance of the timing of birth and indicates that premature transition to the ex utero environment could disturb the well-regulated developmental trajectories of cortical growth, an insight that aligns with literature across various modalities' I don't disagree with the statement, but you did not study this directly?

* Non-singleton participants (79 infants) are referred to as twins in supplementary materials. Why is the number not even, or maybe there were >2 non-singletons? Results on singletons are shown in supplementary, a brief mention in the main text, noting agreement with full sample, would strengthen the results. Otherwise one could argue that non-singletons might not be developing typically due to very different intra-uterine conditions compared to singletons, affecting your developmental relationships. (Similarly, it is very good that you performed the same analysis with the Schaefer atlas too, why are the results not ever mentioned/discussed? Even 1-2 sentences about whether the results agree or not, and why would be helpful.)

* Not sure I agree with the statement suggesting longitudinal studies could 'compensate for the imbalance in postnatal ages'. Longitudinal studies allow observation of trajectories, not correction of GA/PNA differences, or maybe you meant something else?

Minor comments:

Analysis and reporting

* Consider reporting regional variability in intensity distributions across cortical depth as this is interesting in itself.

* It would be nice to get more details on how the intensity values were sampled for the 12 intracortical surfaces (including interpolation).

* Why are results in the main text shown for only one hemisphere? Were left/right regions averaged? Supplementary figures show both.

Terminology, Figures, Legends

* The legend to Fig. 2 may not need to re-explain definitions of the signal descriptors, you provided those in the methods.

* I find it strange that y-axis of the T1w/T2w intensity plots have no indication of 'localisation' even just adding pial,wm would be useful.

* Figure 3: Why were correlations with variance not shown?

* Figure 5A: top right + bottom left and B left + middle - are these exactly same maps and correlations or are they so similar?

* Several figure labels are inconsistent, e.g., "duration of gestation" in Figure 1B is Gestational age? Please double check the terminologies for ages (GA, PMA, PNA) throughout the manuscript and figures.

Other

* Clarify expressions: 'relay weak sensory drive,' 'how translating intracortical microstructure profiling,' 'transfer developmental control from genetic preprogramming to increasing experience-dependent plasticity.'

* 'long-term cognitive vulnerability' preterm infants tend to be affected across multiple domains, including cognitive outcomes, would neurodevelopmental vulnerability be more encompassing?

* I am not sure I understand the 'differentiable' in this sentence: 'In contrast, we observed decreases in profile variance during the postnatal period, suggesting that cortical layers become more differentiable in the first postnatal weeks'. In methods you defined variance as 'the spread of intensities across the profile. A higher variance relates to a flatter intensity profile (i.e. uniformly distributed intensities across intracortical depths) and a low variance describes a heterogeneous intensity distribution across intracortical depths'. Decrease in variance thus reflects more uniform distribution (less differentiable laminae), or maybe I have misunderstood your statement?

* The statement that macroscale features 'do not relate to cytoarchitectural changes' is too strong. While coarse, thickness and surface area reflect some cytoarchitectural aspects (laminar thickness, regional cell density). It would be beneficial to soften this wording

* 'mesiotemporal axis' - 'mesiotemporal lobe' or do you have something more specific in mind?

* 'postnatal age of 6+6 weeks' is this std? +-?

FYI, I tried to provide as much localization in the text as possible for my comments, but without line numbers this was difficult. I would encourage the authors to include them in their future submissions to make the reviewers' lives easier 0: )

Thank you for the opportunity to review this interesting work, I hope my comments will be helpful. With clarification of methodological limitations, careful framing of developmental interpretations, and minor revisions to terminology and figures, the manuscript will provide valuable insight into early cortical microstructural development. I wish the authors the best in their revisions.

Reviewer #2: The manuscript provides a thorough and rigorous analysis of T1/T2 ratio offering insights into the spatial patterning of microstructure following birth. The results are complemented by a thoughtful summary of early biological mechanisms governing areal patterning of the human cortex. However, the interpretation of the changes in the context of sensory isolation vs external sensory stimulation from the postnatal environment is a bit oversimplistic. The text would be improved with added nuance to the interpretation of findings.

"After birth, aside from the intrinsic refinement of the cortical structure, sensory input drives neuronal patterning." Neonates spend approximately 70% of their time asleep. Given the disproportionate amount of time spent asleep, research suggests that electrical activity during sleep plays a crucial role in the developing motor and visual system (PMCID: PMC4884844; PMCID: PMC9254005).

"In utero the foetus is relatively isolated from sensory input and therefore genetic and epigenetic programming primarily steer neurobiological processes" There is sensory input from maternal voice and heartbeat (PMCID: PMC4364233) and sensory input from motor movements from the infant during sleep (PMID: 26364005) that likely contribute to brain development.

"This modulatory transition highlights the importance of the timing of birth in brain development, with prematurely born infants being exposed to a sensory-rich environment before the scaffold of their circuitry has been completely established for postnatal plasticity." The clinical characteristics of the preterm neonates included in the analysis are not described. However, the study includes extremely preterm and very preterm neonates. Extremely preterm and very preterm neonates often have interventions to sustain life (e.g., mechanical ventilation, antenatal steroids, humidity and oxygenation control, antibiotics). Disruptions to oxygenation and blood flow, and associated brain injury, are more likely the culprit to aberrant grey and white matter development than a 'sensory-rich environment'. Further, disrupted sleep from feeding and for necessary medical interventions is likely a major contributing factor to disrupted brain growth in prematurely born infants.

"In general, the prominent effect of gestational maturation on cortical microstructure emphasises the importance of the timing of birth and indicates that premature transition to the ex utero environment could disturb the well-regulated developmental trajectories of cortical growth, an insight that aligns with literature across various modalities" It's not clear that the current results demonstrate that premature birth disturbs developmental trajectories. Is there an interaction between postnatal age and GA? i.e., do neonates born extremely or very preterm have a lower postnatal correlation with microstructure intensity than term born neonates?

Minor

The abstract says (n=599, 0-7 weeks). Rather than the postnatal age, it would be more informative to state the GA or PMA.

The x-axis is extremely small font in figures 1-2.

Reviewer #3: This manuscript investigates how intrinsic and extrinsic factors drive maturation of the human cortex during late gestation and early postnatal development. The authors used T1-weighted and T2-weighted MRI scans from the Developing Human Connectome Project (dHCP), leveraging the T1w/T2w ratio as a biomarker to characterize cortical microstructural changes. To present intracortical microstructure profiles, the neonatal cortex was divided into 12 layers, and the profile shapes were described using central moments inspired by prior histological work. The effects of gestational and postnatal age on these profiles were examined and found to align with low spatial-frequency geometric eigenmodes of the cortex. It is commendable that the authors attempted laminar-level analysis of the neonatal cortex to assess intrinsic and extrinsic factors, and the relatively large dataset of 599 neonates is a notable strength. However, without a strong fundamental basis for dividing the neonatal cortex into 12 layers, and without sufficient justification of the accuracy of values extracted from each layer, the subsequent analyses risk being perceived as a mathematical exercise rather than a substantive contribution to our understanding of cortical development. Such analysis to MRI is fundamentally limited by much lower resolution of MRI compared to that of histological images. It is almost impossible to justify expanding histological framework to MRI data as conducted in this manuscript.

Major comments:

1. The authors constructed 12 equivolumetric surfaces between the pial and white matter surfaces. However, the average human cortex is approximately 2-4 mm thick, and neonatal cortices are even thinner. The raw T1-weighted image resolution is 0.8 x 0.8 x 1.6 mm3. In theory, the images can be upsampled to any higher resolution, but any upsampling is just artificial, not bringing real new information. Dividing the neonatal cortex with acquisition resolution of only 1-2 voxels into 12 layers appears to be just mathematic exercise instead of revealing any validated information. The cited paper (Waehnert et al., 2014) conducted the analysis on ex-vivo images acquired on 7T scanner with resolution of 140 um, which cannot fully justify the same methods could be employed in this study. In the current context, this approach raises concerns about artificial measures, and a high degree of voxel overlap between adjacent layers. Furthermore, the segmentation results shown in Fig. 2A appear incorrect in the central regions of the brain, which raises additional concerns about segmentation accuracy even before performing laminar analysis. The partial volume effects are big. The deepest cortical layers are likely to be contaminated by white matter voxels, while the most superficial layers may be affected by CSF signals. Finally, no laminar surface mapping figures were presented in the main results or supplementary materials, which further undermines validity in the study's central claim of capturing cortical laminar organization.

2. While one of the major highlights in the Methods is "histology-inspired intracortical profiling", it is not clear how the measure -- center of gravity (CoG) was computed in this study, and the cited reference (Schleicher et al., 1999) does not provide a clear corresponding explanation. If this is the first time this measure was employed in the imaging study, more detailed and rigorous explanation or testing of the measure is expected. Figure 2B is particularly confusing: of all three small panels, the x-axis is labeled as T1w/T2w intensity and the y-axis as cortical depth, but it is unclear how the different panels separately represent mean intensity, CoG, and variance. In addition, the meaning of the color coding is not explained—does it represent age or these center measure?

Minor comments:

1. The authors also conducted product-moment correlations between the postmenstrual, gestational and postnatal age effects and the 2nd-4th geometric eigenmodes. However, the manuscript does not provide a detailed explanation of how these eigenmodes were computed, and the cited reference (Pang et al., 2023) is missing from the reference list.

2. Throughout the manuscript, terms such as "microstructural density," "intracortical homogeneity," and "variance" are sometimes used interchangeably or without precise definition. For readers less familiar with histology-inspired moment metrics, a more consistent phrasing would make the results easier to interpret.

---

## [Editor Report · Decision Letter 2]

23 Feb 2026

Dear Dr Paquola,

Thank you for your patience while we considered your revised manuscript "Disentangling the influences of pre- and postnatal periods on human cortical myelin" for publication as a Research Article at PLOS Biology. This revised version of your manuscript has been evaluated by the PLOS Biology editors and the Academic Editor.

Based on our Academic Editor's assessment of your revision, we are likely to accept this manuscript for publication, provided you satisfactorily address the following data and other policy-related requests:

* We would like to suggest a different title to improve its accessibility for our broad audience:

Prenatal and early postnatal periods differentially shape the maturation of human cortical microstructure and myelin

* Please add the links to the funding agencies in the Financial Disclosure statement in the manuscript details.

* Please include the approval/license number of the ethical approval for the experiments.

* Please include information in the Methods section whether the study has been conducted according to the principles expressed in the Declaration of Helsinki.

* Please specify whether the participants provided written or oral consent.

* DATA POLICY:

Regardless of the method selected, please ensure that you provide the individual numerical values that underlie the summary data displayed in the following figure panels as they are essential for readers to assess your analysis and to reproduce it: 5B

* CODE POLICY

Per journal policy, if you have generated any custom code during the course of this investigation, please make it available without restrictions. Please ensure that the code is sufficiently well documented and reusable, and that your Data Statement in the Editorial Manager submission system accurately describes where your code can be found. More information on our Code Policy, what and how to share can be found here: https://journals.plos.org/plosbiology/s/code-availability

We expect to receive your revised manuscript within two weeks.

*Published Peer Review History*

*Press*

Sincerely,

Christian

Christian Schnell, PhD

Senior Editor

cschnell@plos.org

PLOS Biology

---

## [Editor Report · Decision Letter 3]

9 Mar 2026

Dear Dr Paquola,

Thank you for the submission of your revised Research Article "Prenatal and early postnatal periods differentially shape the maturation of human cortical microstructure and myelin" for publication in PLOS Biology. On behalf of my colleagues and the Academic Editor, Henry Kennedy, I am pleased to say that we can in principle accept your manuscript for publication, provided you address any remaining formatting and reporting issues. These will be detailed in an email you should receive within 2-3 business days from our colleagues in the journal operations team; no action is required from you until then. Please note that we will not be able to formally accept your manuscript and schedule it for publication until you have completed any requested changes.

While you attend to those requests to come, please also provide a link in the figure legend of Figure 5 where the source data for this figure can be found, for example: Data used to generate this figure can be found at https://doi.org/10.5281/zenodo.18910238."

PRESS

Sincerely,

Christian

Christian Schnell, PhD

Senior Editor

PLOS Biology

cschnell@plos.org